# Using the Behaviour Change Wheel to develop an intervention to improve conversations about recovery on the stroke unit

Louisa-Jane Burton[1,2]*, Anne Forster[1,2], Judith Johnson[3,4¤], Thomas F. Crocker[1,2], David J. Clarke[1,2]

1 Academic Unit for Ageing & Stroke Research, Bradford Teaching Hospitals NHS Foundation Trust, Bradford, United Kingdom, 2 Academic Unit for Ageing & Stroke Research, Leeds Institute of Health Sciences, University of Leeds, Leeds, United Kingdom, 3 School of Psychology, University of Leeds, Leeds, United Kingdom, 4 School of Public Health and Community Medicine, University of New South Wales, Sydney, Australia

¤ Current address: Division of Nursing, Midwifery and Social Work, School of Health Sciences, University of Manchester, Manchester, United Kingdom
* louisa.burton@bthft.nhs.uk

**Data Availability Statement:** The data underlying the survey results presented in the study are available from: https://osf.io/9vyxd. There are some

## Abstract

### Background

Understanding recovery is important for patients with stroke and their families, including how much recovery is expected and how long it might take. These conversations can however be uncomfortable for stroke unit staff, particularly when they involve breaking bad news. This study aimed to begin development of a novel complex intervention to improve conversations about recovery on stroke units.

### Methods

Informed by previously collected qualitative data, we used the Behaviour Change Wheel (BCW) approach to identify possible 1. barriers to communication about recovery on stroke units; 2. Intervention Functions; 3. Behaviour Change Techniques (BCTs) to incorporate in an intervention. We subsequently sought stroke professionals' perspectives through an online survey. Respondents rated the importance of barriers for intervention inclusion and evaluated the usefulness and feasibility of the suggested BCTs.

### Results

Our behavioural diagnosis identified a target behaviour of provision of information about recovery by stroke unit professionals to patients and carers. Twelve possible barriers to this behaviour were identified, with six potential Intervention Functions to address them, and 29 BCTs. Forty-eight multidisciplinary professionals responded to the survey. The six barriers rated as most important to address were: lack of confidence; perceptions of insufficient communication skills; lack of knowledge of the benefits; difficulties in deciding when and in what

restrictions on public access to data from the ethnographic study which informed this research. These restrictions relate to the consent provided by the participants in the study for access to data they provided for the study (approved by the Health Research Authority (HRA) for the United Kingdom (UK) and the Research Ethics Committee (REC) (Yorkshire & the Humber (Bradford-Leeds). Participants did not provide informed consent for their data to be made available and these data contain potentially identifying or sensitive patient information.

**Funding:** Funding for this work was provided by the Stroke Association, via its Postgraduate Fellowship programme [ref: TSA PGF 2017-02].

format to provide information; absence of private spaces for discussions; and lack of generic written information to support conversations. The developed intervention strategy comprised twelve clinically feasible and useful BCTs, encompassing the Intervention Functions of Training, Enablement, Persuasion, and Environmental restructuring.

## Conclusions

The BCW approach was successfully used to begin development of an intervention to improve conversations about recovery on the stroke unit; our survey enabled incorporation of stakeholder perspectives. Further development work is required to design intervention materials and test whether the strategies are effective in improving staff and patient outcomes.

## Introduction

It is estimated that one in four people aged over 25 will have a stroke in their lifetime [1]. Although enhanced acute treatments are improving survival rates [2] and some recovery is nearly always possible, stroke remains one of the leading causes of disability [3], with the potential to impact physical functioning and cause a range of 'hidden' deficits, including psychological and communication difficulties, and fatigue [4, 5]. Hospital-based rehabilitation aims to increase survivors' independence and maximise their quality of life, however around one in three people are discharged requiring help with daily activities [6] and a tenth require subsequent institutional care [7].

Understanding both how recovery occurs and the likely progress they might make is important for patients and their families (many of whom subsequently take on additional caring responsibilities). This information can help them to adjust to any ongoing disabilities [8, 9], and engage in shared decision-making [10, 11], in preparation for life after stroke. Around a third of patients and carers however report that hospital-based rehabilitation staff (including doctors, nurses, and other allied health professionals) do not do enough to help support their understanding [12] and information about recovery is frequently described as an unmet need [13].

A growing body of evidence has highlighted the challenges faced by healthcare professionals when sharing information about post-stroke recovery with patients and carers, particularly when discussing the potential for ongoing disability [9, 14–16]. These include difficulties in predicting and sharing the often-uncertain trajectory, the involvement of a range of multidisciplinary professionals, and concerns about eroding patients' hope and motivation in therapy, often perceived by professionals to subsequently lead to inferior outcomes [14, 17]. Additionally, staff report a lack of training, particularly in breaking bad news [14–16]. Many receive only basic communication skills training within their professional education programmes and are required to learn advanced skills experientially [14, 16]. This can result in a lack of confidence and psychological stress when faced with providing information to patients and carers and managing their responses [14, 15].

Despite the recognised challenges, no interventions specifically designed to address the problem of communication of post-stroke recovery information could be identified. Although training to improve communication skills (including in breaking bad news) is available for staff in other clinical areas, e.g., palliative or cancer care, these interventions often lack

theoretical grounding [18] and do not address the challenges specific to stroke, which include the need to support continued engagement in therapy and the involvement of a multidisciplinary team of professionals [16]. Development of such interventions has the potential to improve staff confidence and skills, and could lead to enhanced patient and carer involvement in planning and decision-making, and adjustment. Informed by the Medical Research Council (MRC) Framework for developing and evaluating complex interventions [19], we began development of a novel complex intervention to improve conversations about recovery on stroke units.

A core component of the MRC framework is the use of theory. Complex interventions require recipients to change their behaviour in some way; thus developing effective interventions requires understanding of the underlying psychological mechanisms that drive this behaviour change, and how they operate [20]. In practice, this understanding guides the selection of techniques and components within an intervention, which are theorised to modify these constructs and therefore result in behaviour change [21]. In this study, we used the Behaviour Change Wheel (BCW) approach to guide our early intervention development work [22, 23]. The approach is based on a synthesis of the features of nineteen behaviour change frameworks [23]. At its centre lies the Capability, Opportunity, Motivation model of Behaviour (COM-B), which posits that for an individual to exhibit any behaviour, they must have the capability and the opportunity, and be motivated to do so, more than any other or no behaviour [23]. These interacting components (or combinations of them) represent the sources of behaviour that could potentially be targeted by an intervention [22]. A step-by-step guide to using the BCW approach to develop interventions has been developed by Michie and colleagues, which involves developing an understanding of the behaviour using the COM-B model; identifying interventions options; and developing intervention content and implementation strategies [22]. This approach has been widely applied, including to develop stroke rehabilitation interventions targeting the reduction of sedentary behaviour [24], and increased in upper limb exercise [25] and active practice [26], as well as strengthening the role and functions of nurses [27].

In this study, we aimed to begin development of a novel intervention by identifying theory-based and clinically feasible intervention strategies to improve conversations about recovery on the stroke unit. Our objectives were to:

1. Use behaviour change theory (the BCW approach [22]), informed by previously collected qualitative data, to identify and understand the behaviour (including the barriers to changing it), and identify potential Intervention Functions and Behaviour Change Techniques (BCTs) to address the identified barriers;

2. Conduct an online survey to engage stakeholders (stroke unit professionals) to assess the validity of, and prioritise, the identified barriers to change, support selection of BCTs, and identify options for mode of delivery.

## Materials and methods

We used a two-phase approach to intervention development, reported in accordance with the Guidance for Reporting Intervention Development Studies in Health Research (GUIDED) checklist (see S1 File). In Phase 1, we addressed Objective 1, working through the BCW approach to select and understand the behaviour, identify (informed by previously collected qualitative data) barriers to it, and select potential Intervention Functions and Behaviour Change Techniques (BCTs) to address these barriers. In Phase 2 (addressing Objective 2), we

used an online survey to present the identified barriers to stakeholders (stroke unit professionals), and asked them to assess and share their thoughts on the importance of addressing each one in an intervention, and their perceptions of the usefulness and clinical feasibility of the proposed BCTs, and how these could be delivered.

## Phase 1: Application of the BCW approach

In the first phase, the lead author (LB, an experienced qualitative researcher) initially worked through the BCW approach. Methods and findings were regularly discussed across the multidisciplinary research team, which included researchers with backgrounds in nursing (DJC), physiotherapy (AF) and clinical psychology (JJ), all of whom had experience of clinical work with stroke survivors.

The first stage in the BCW approach is to understand the behaviour [22]. Through four steps, intervention developers are directed to 1) define the problem in behavioural terms; 2) identify the behaviour to target with an intervention; 3) specify the behaviour and; 4) identify what needs to change to enable the desired behaviour to be realised, categorised using the COM-B model [22].

Michie et al. recommend collection of primary data to understand the behaviour and what needs to change from multiple stakeholder perspectives, with triangulation of data from a range of sources to increase confidence [22]. In this study, previously collected qualitative data from a focused ethnographic case-study in two UK stroke units were utilised, including observations (N = 84), interviews with patients (n = 10), carers (n = 4) and professionals (n = 19), and documentary analysis of written records (these data, alongside more detailed methods, are reported separately [16]). This data collection was influenced by the COM-B model: interview topic guides included questions about the barriers to discussions about recovery, with prompts around Capability, Opportunity and Motivation. Ethical approval for this work was provided by the Health Research Authority (Yorkshire & the Humber (Bradford-Leeds) NHS Research Ethics Committee Ref 19/YH/0009). Participants deemed to have capacity to consent provided written informed consent; a written consultee declaration was sought for those who lacked capacity (these participants took part in observations and documentary analysis only). Recruitment took place between 11/03/2019-31/12/2019.

Findings from this qualitative work informed selection of potential target behaviours (which were discussed amongst the research team), including consideration of the potential impact and likelihood of changing the behaviour, how changes could be measured and whether such behaviour changes would result in any 'spillover' effects to other behaviours [22]. To identify what needed to change, qualitative data about the potential factors (barriers and facilitators) influencing conversations about recovery were extracted and mapped to the COM-B model, to identify professionals' Capability (physical and psychological) to perform the target behaviour, Opportunity (physical and social) for the behaviour to occur and their Motivation (automatic and reflective) to engage in the behaviour. Michie et al. present an additional optional step in utilising the Theoretical Domains Framework (TDF) [28] to develop more detailed understanding of the behaviour [22]. Incorporating a range of behaviour change theories, the TDF consists of fourteen domains, which represent theoretical constructs, each linked to COM-B components [28]. In this study, we chose to use the TDF alongside the COM-B to further classify and understand what needed to change; the TDF encouraged us to consider a broader range of influences (barriers and facilitators) on professionals' behaviour [29] and aided later selection of BCTs linked to these domains. Links between the COM-B components and TDF domains are detailed in Fig 1. Initial familiarisation with the data was followed by the coding of data (interview transcripts, observational fieldnotes, documentary

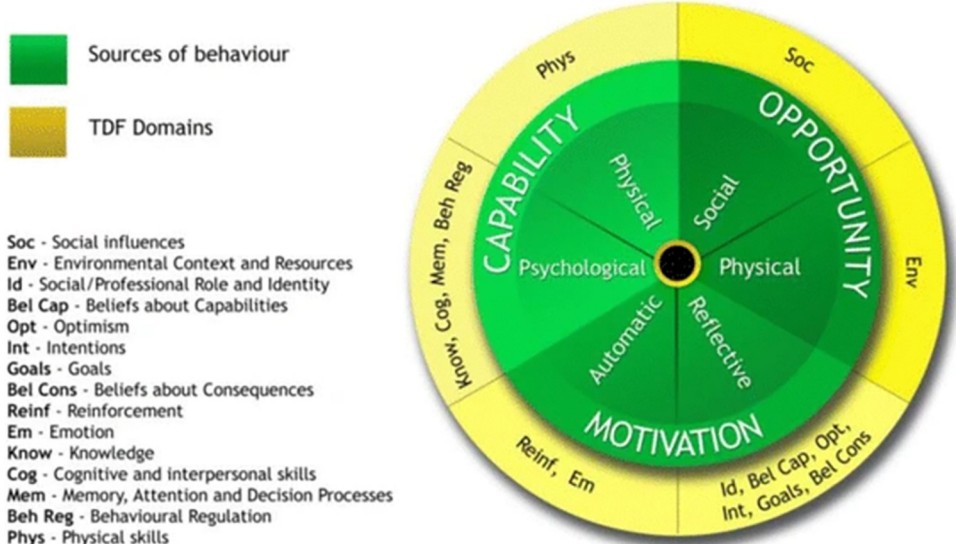

**Fig 1. The relationship between the COM-B model and TDF domains.** Reproduced from Atkins et al., 2017 (Fig 1, pp 11 [30]). TDF = Theoretical Domains Framework. Licensed under CC-BY 4.0: see https://link.springer.com/article/10.1186/s13012-017-0605-9#rightslink.

data) highlighting factors with the potential to influence conversations about recovery in QSR NVivo (v12). LB initially categorised these influencing factors according to the COM-B domains, then using the TDF to further classify them and prompt the identification of additional factors. Through an iterative process, this behavioural analysis was reviewed and revised through discussion with the research team.

In the second stage of the BCW approach, the nine Intervention Functions suggested by Michie et al. were considered. LB devised a comprehensive list of all Intervention Functions, which were linked to each of the TDF domains in which barriers had been identified, and so could theoretically be effective in modifying the target behaviour. Michie et al. suggest considering these as options and encourage intervention developers to be confident in using their own judgement, alongside the APEASE criteria to select those most appropriate to the context [22]. Using the APEASE criteria requires consideration of the Affordability, Practicability, Effectiveness and cost-effectiveness, Acceptability, Side-effects/ safety, and Equity (the APEASE criteria) [22]. In this context, affordability and practicability were prioritised due to resource limitations in UK healthcare settings. LB initially judged the options for Intervention Functions in line with the APEASE criteria, before discussing these judgements with the wider team.

Michie et al. suggest seven policy categories, which may be effective and appropriate in delivering the selected intervention functions [22]. LB again reviewed these alongside the APEASE criteria, identifying which were appropriate to the context in which the intervention was intended to be implemented before they were reviewed and agreed upon by the wider research team.

In the final stage of the BCW approach, the selected Intervention Functions were linked to more specific BCTs [22]. The APEASE criteria were considered to make judgements about which were most likely to be appropriate to consider in the context of developing an intervention to improve provision of post-stroke recovery information. This work was undertaken by LB in the first instance; the research team then reviewed this list of BCTs alongside the identified barriers and intervention functions and revisions took place through discussion.

At this point, the number of potential BCTs remained high. Although BCTs had been selected with consideration of the APEASE criteria, we were conscious that these decisions relied upon the subjective judgements of the research team. Although some members had previous clinical experience, and we used our previous qualitative findings to inform our decision-making, we felt this was an important point to seek the views of stakeholders (stroke unit professionals). This is in line with MRC guidance recommending the involvement of stakeholders throughout the intervention development process [19].

## Phase 2: Online survey study

In Phase 2, we conducted an online survey of stroke unit professionals to investigate the importance of addressing the identified barriers in an intervention (thus examining their validity in a wider range of stroke units than the local ethnographic work upon which they were originally based), and the feasibility and usefulness of the identified BCTs (and how they might best be delivered), with the aim of selecting those most likely to be acceptable to professionals and feasibly implemented (Objective 2). An online survey was developed using JISC Online Surveys. Questions were drafted by LB and reviewed by the research team, who suggested amendments prior to piloting. In the pilot stage, seven researchers with previous professional stroke unit experience completed the questionnaire online alongside a form asking them to comment on: ease of navigation; understanding of the language used; clarity of the instructions; question order; and survey length. Feedback was used to refine the questionnaire; changes included re-ordering some of the questions and minor clarifications to language. The questionnaire (available at https://osf.io/rxqks) collected demographic information (age group, gender, profession, and time working in stroke care), before presenting each identified barrier and the BCTs suggested to address it. Fixed-choice questions using Likert scales asked about the importance of addressing the barrier in an intervention (from 1 = not (at all) important to 5 = extremely important) and, for each suggested BCT, how likely to be (a) feasible and (b) useful to address the barrier in question (from 1 = very unlikely to be feasible/ useful to 4 = very likely to be feasible/ useful). Free text space was provided to enable participants to expand on their answers and suggest alternative strategies.

Ethical approval was received from the School of Medicine Research Ethics Committee, University of Leeds (ref: MREC 21–013); informed consent was collected as part of the online survey form. Recruitment took place between 14/02/2022-14/06/2022. Inclusion criteria were broad, with participants self-identifying as a qualified healthcare professional of any discipline currently working in a UK stroke unit. The survey was advertised online (via X (Twitter) and the Stroke Network online forum of the FutureNHS Collaboration platform) and via the professional bodies of potential participants (including the National Stroke Nursing Forum, Organisation for Psychological Research into Stroke, Royal College of Occupational Therapists and Royal College of Speech & Language Therapists). Potential participants followed a link from one of these sources, which presented information about the study and a downloadable participant information sheet. Informed consent and questionnaire responses were provided through the online survey; participants created a unique code enabling them to withdraw following submission if they wished.

Quantitative data were managed in SPSS Statistics and analysed descriptively (anonymised data available from: https://osf.io/9vyxd). Median participant ratings of the importance of each barrier were calculated, as was the percentage of participants who rated each barrier as either very or extremely important to address in an intervention. Subsequently, the percentage of participants who rated each BCT as likely to be useful or very useful, and feasible or very feasible was calculated.

An exploratory approach was undertaken to select the BCTs most likely to be useful and feasible. Firstly, barriers were ranked according to the percentage of participants who rated them as very/ extremely important to address in an intervention (highest to lowest), to identify the five most important. We chose to select the five most important barriers as a pragmatic step to focus the intervention on what stakeholders considered most important, and to ensure that it did not become so complex such that it became unmanageable and could not be implemented. The BCTs selected to address each of these barriers were compiled, which were first ranked according to the percentage of participants who rated them as likely or very likely to be useful, with the twenty highest ranking BCTs selected. Again, the decision to select the twenty highest ranking BCTs was a pragmatic one, designed to limit the complexity of the intervention. The remaining BCTs were ranked again, according to the percentage of participants rating them as likely or very likely to be feasible. Those rated as least likely to be feasible were removed, with the remainder retained for intervention inclusion.

Qualitative data gathered from free text responses were subject to directed content analysis [31]. Responses were imported into Microsoft Excel. LB initially read through all the comments related to each question, then coded them according to the study objectives (i.e., whether they related to the importance of addressing a barrier; perceived usefulness of a suggested BCT; perceived feasibility of a suggested BCT; or an alternative strategy to address the barrier). Descriptive summaries of participants' responses were then developed, which were reviewed and discussed amongst the research team.

Finally, as a research team, we considered mode of delivery, with options including face-to-face or distanced delivery, and individual or group approaches, selected using the APEASE criteria [22] and informed by qualitative responses from the survey.

## Results

### Phase 1: Application of the BCW approach

Our previous qualitative work identified patients' and carers' perceptions that information was not provided proactively, highlighted inequity in the opportunities offered to discuss it, and suggested provision of unclear or inconsistent information from different professionals. Therefore, the problem was defined in behavioural terms as "Patients and families do not receive adequate information about recovery after stroke." Two potential target behaviours were considered. Firstly, consideration was given to designing an intervention targeting the behaviour of patients and carers, i.e., encouraging and empowering them to seek out information about recovery from stroke unit professionals. Although it was felt likely that patients' and carers' behaviour could be changed, and that this behaviour could be measured (e.g., via the number of requests for information), the potential impact of changing this behaviour, and potential spillover effects to professionals' behaviour, were felt to make it unpromising. For example, our previous qualitative work [32] revealed a range of issues experienced by professionals, including difficulty in predicting recovery in some situations, and the emotional cost to themselves, which might render them unprepared to manage requests for information and the emotionally challenging conversations that might ensue. Targeting professionals' behaviour was felt to be more promising, both in the impact and likelihood of behaviour change. Additionally, the potential for 'spillover' effects was identified, in which the culture of the unit could change, with discussions about recovery regularly taking place, thus potentially empowering patients to seek information more readily. Therefore, the selected target behaviour focused on staff and was "Providing information about recovery by stroke unit professionals to patients and their carers." The behaviour is specified more precisely in Table 1.

**Table 1. Specifying the target behaviour.**

| | |
|---|---|
| **What?** | Providing information to patients and families, including both generic information about recovery process and personalised information about the patients' likely progress, in an appropriate format to meet the information recipient's needs, (likely to be primarily through conversation, but supplemented with written (including accessible) materials to meet patients' and carers' needs) |
| **Who?** | As qualitative work highlighted professionals' perceptions that provision of information about recovery was the responsibility of all staff, the people involved in the behaviour are defined as multidisciplinary stroke unit staff, including both qualified professionals (e.g., doctors, nurses, therapists) and other staff members (e.g., therapy and healthcare assistants, housekeepers, ward clerks, porters). |
| **With whom?** | Information should be provided to patients (where appropriate to their level of understanding and their wishes) and/ or their carers (with the patient's express permission where they are able to provide it or in their best interests if not). |
| **When?** | Qualitative work identified that at least some information could be provided during the in-patient stay (even when this was short). Although the type of information may change over the course of the hospitalisation, e.g., generic information is more likely to be provided in the acute phase, before individual assessments and multidisciplinary discussions have taken place; there is a role for regular information provision from admission, through to discharge. |
| **How often?** | Information should be provided as often as is required, but at a minimum, information should be offered at least once during the hospital admission. |
| **Where?** | On the stroke unit/ therapy unit or other location in the hospital, preferably in a quiet, private, and accessible area. |

Twelve barriers (areas where change was potentially required to improve provision of information about recovery) were identified and classified (detailed in Table 2). Overall, the analysis revealed four components of the COM-B model where changes were potentially needed: Psychological Capability (five barriers), Physical Opportunity (two barriers), Reflective Motivation (four barriers) and Automatic Motivation (one barrier). These barriers were classified into eleven TDF domains: Knowledge; Cognitive and interpersonal skills (two barriers); Memory, attention and decision processes; Behavioural regulation; Environmental context and resources; Professional/ social role and identity; Beliefs about capabilities; Beliefs about consequences (two barriers); and Emotion.

All nine Intervention Functions were linked to the identified COM-B components and TDF domains and considered as options. Following application of the APEASE criteria, the six Intervention Functions selected as most appropriate for the context in which behaviour change was required were: Education, Persuasion, Training, Environmental restructuring, Modelling and Enablement. Although Incentivisation and Coercion were linked to the component of Automatic Motivation, consideration of the APEASE criteria led to the judgements that creating expectations of reward or punishment would be impractical and potentially unacceptable in this context. Restriction (using rules to increase the target behaviour by reducing the opportunity to engage in competing behaviours) was linked to Physical Opportunity but was also considered impractical.

Two policy categories were established as being potentially useful in this context: Guidelines and Service provision. The 2023 National Clinical Guideline for Stroke is particularly pertinent, with new recommendations in the latest edition recommending that information about functional prognosis and likelihood of goal achievement is shared to manage patients' expectations [33]. This may act as an enabler for intervention implementation. Service provision was also highlighted as an enabler, as the delivery of information about recovery to patients and carers must take place in the context of the service provided in the stroke unit. Other policy options were considered impracticable within the context of this intervention.

At this stage, our list comprised 29 potentially relevant individual BCTs (with some used to address multiple barriers; see Table 2). This was deemed too many to combine into a single

**Table 2. Behavioural diagnosis using the COM-B model, suggested intervention functions and associated BCTs.**

| COM-B component | Theoretical Domains Framework domain | Relevance of domain (facilitators to performing behaviour) | Barriers (what needs to change) | Intervention functions | Individual BCTs |
|---|---|---|---|---|---|
| Psychological capability | Knowledge | Professionals have the knowledge to make predictions about individual patients' recovery | Some professionals (particularly junior staff) describe feeling unable to predict recovery | Training | Provide information on factors to consider, which may impact recovery after stroke (**Instruction on how to perform a behaviour**) |
| | Cognitive and interpersonal skills | Professionals possess the required communication skills to deliver information about recovery sensitively and compassionately | Some professionals perceive that they do not have the required communication skills to deliver information about recovery, particularly when this involves breaking bad news | Training | Provide instruction about how to discuss recovery sensitively and compassionately (**Instruction on how to perform a behaviour**) Demonstrate how to deliver information about recovery sensitively and compassionately (**Demonstration of the behaviour**) Prompt practice of conversations about recovery through role play with peers (**Behavioural practice/ rehearsal**) Provide feedback following observation of practice conversations with peers (**Feedback on behaviour**) |
| | | Professionals are able to assess whether and how much information patients and families want to know about recovery | Some professionals may find it difficult to assess whether and how much information about recovery to provide to individual patients and few report directly asking patients and families about how much information they would like to receive | Training | Advise on how to ask patients and carers about whether and how much information about recovery they wish to receive (**Instruction on how to perform a behaviour**) Demonstrate how to ask patients and carers about whether and how much information about recovery they wish to receive (**Demonstration of the behaviour**) Prompt practice of asking how, whether and how much information is wanted through role play with peers (**Behavioural practice/ rehearsal**) Provide feedback following observation of practice conversations with peers (**Feedback on behaviour**) |
| | Memory, attention, and decision processes | Professionals are able to decide when and in what format to provide information to individual patients/ families | Some professionals find it difficult to decide when and in what format to provide information about recovery to meet individual patients' needs, e.g., where patients have cognitive or communication problems | Training, enablement | Advise on how to decide when and in what format to provide information to individual patients/ families (**Instruction on how to perform a behaviour**) Demonstrate examples of conversations occurring in different ways (e.g., at different times, supported by written documentation or not) with patients with different needs (**Demonstration of the behaviour**) Prompt practice of making decisions about when and in what format to provide information to individual patients/ families using vignettes (**Behavioural practice/ rehearsal**) Provide feedback following discussions based on vignettes (**Feedback on behaviour**) Advise on requesting support from colleagues across the MDT when making decisions about when and in what format to provide information (**Social support (practical)**) |
| | Behavioural regulation | Standard procedures are in place to monitor whether, when and to whom information has been provided, to promote consistency across patients | Professionals do not routinely provide information about recovery to all patients, potentially resulting in inequity | Education, Training, Enablement | Encourage a unit-specific plan to provide information about recovery, e.g., at specific time-points/ in specific contexts (**Action planning**) Prompt conversations about recovery at specific time-points during admission, e.g., every 2 weeks (**Prompts/cues**) Agree on a goal of having a conversation about recovery with all patients/ families at certain timepoints in their admission, e.g., every two weeks (**Goal-setting behaviour**)). Examine how performance fits with agreed goal (through audit) and consider modification if needed (**Review behaviour goals**). Put in place physical reminders (e.g., in patients' records), or verbal prompts (e.g., at MDT meetings) to alert professionals at the time when a conversation about recovery is due (**Prompts/ cues**) Establish a single shared record for the MDT to monitor whether information has been provided and record the outcome of conversations (**Self-monitoring of behaviour**) Arrange for professionals to remind each other about providing information, e.g., through regular prompting at formal meetings or informal supervision (**Social support (practical)**) Prompt professionals to identify barriers when conversations about recovery have not taken place and discuss ways to overcome them as a team (**Problem-solving**) |

*(Continued)*

**Table 2.** (Continued)

| COM-B component | Theoretical Domains Framework domain | Relevance of domain (facilitators to performing behaviour) | Barriers (what needs to change) | Intervention functions | Individual BCTs |
|---|---|---|---|---|---|
| **Physical opportunity** | Environmental context and resources | Quiet and private spaces to provide information about recovery are available on the ward, to promote confidence and facilitate patients' and families' receipt of the information | Professionals may lack opportunities to provide information about recovery due to the absence of appropriate private and quiet spaces to speak with patients/ families, felt to be necessary for patient confidentiality and to support receipt of the information, e.g., noise/ distractions can result in difficulties taking in information | Training, Environmental restructuring, Enablement | Advise on importance of providing information about recovery in a private and quiet area, and how to prevent interruptions **(Instruction of how to perform a behaviour)** Advise on allocation of designated areas as quiet and private areas to discuss recovery **(Restructuring the physical environment)** |
| | | Written generic information about recovery is available for professionals to use to support conversations about recovery with patients and families | Little written information about recovery is available for professionals to support conversations, particularly for patients/ families with cognition or communication problems | Training, Environmental restructuring, Enablement | Provide (or support professionals to identify) generic written information to provide to patients/ families **(Adding objects to the environment)** Ensure written information is readily available in a specific location for professionals to access when required **(Prompts/ cues)** |
| **Social opportunity** | Social influences | N/A | N/A | N/A | N/A |
| **Reflective motivation** | Professional/ social role and identity | Professionals understand their own and their colleagues' professional roles in providing information about recovery to patients and families | Some professionals, e.g., nurses, may not view discussing recovery as part of their role. | Education, Persuasion, Modelling | Tell professionals that other members of the MDT appreciate their contributions to provision of information about recovery **(Information about others' approval)** Present communication by someone senior with each profession about the importance of talking about recovery as part of their professional role **(Credible source)** Inform the professional that if they provide information about recovery, this will set a good example to other members of their discipline **(Identification of self as role model)** Provide examples of the roles and responsibilities of each professional and the team in providing information for them to aspire to **(Demonstration of the behaviour)** |
| | Beliefs about capabilities | Professionals feel confident in their ability to share information about recovery with patients and families | Professionals (particularly junior staff) report a lack of confidence in sharing information about recovery with patients and families, which may lead them to avoid providing information or providing vague information | Persuasion, enablement | Tell the professional they have the skills and experience to successfully share information about recovery with patients and families **(Verbal persuasion about capability)** Encourage professionals to think about times they have successfully shared information with patients and families and information was well-received **(Focus on past success)** Advise professionals to imagine discussing recovery with patients and families and the information being well-received **(Mental rehearsal of successful performance)** Encourage professionals to provide support and encourage their colleagues when they have had discussions with patients and families about recovery **(Social support (unspecified)** |
| | Optimism | N/A | N/A | N/A | N/A |

*(Continued)*

**Table 2.** (Continued)

| COM-B component | Theoretical Domains Framework domain | Relevance of domain (facilitators to performing behaviour) | Barriers (what needs to change) | Intervention functions | Individual BCTs |
|---|---|---|---|---|---|
| | Beliefs about consequences | Professionals believe that providing information about recovery provides benefits to patients and families and unaware of the risks of not providing such information | Some professionals are unaware of the benefits of providing recovery information to patients and families (e.g., making future plans or adjusting to life post-stroke), and risks to not providing information, (e.g., limiting ability to plan, preventing adjustment) | Education, Persuasion | Provide information on patients' and carers' information needs about recovery from established literature (**Information about social and environmental consequences**) Present a speech by an expert (researcher or professional) outlining the known benefits and risks to providing information about recovery (**Credible source**) Present a speech by an expert (stroke survivor or carer) outlining the known benefits and risks to providing information about recovery (**Credible source**) Provide information about the emotional consequences for patients and carers if information about recovery is not provided effectively (**Information about emotional consequences**) Ask professionals to try providing information about recovery (after structured training and as part of supervised practice) and to note patients' and families' reactions (**Behavioural experiments**) Encourage audit of patient and family feedback about the benefits and disadvantages of providing information about recovery (**Feedback on outcome(s) of behaviour**) Encourage professionals to ask patients and families about the benefits of receiving information about recovery and the problems with not receiving such information (**Self-monitoring of outcome(s) of behaviour**) |
| | | Professionals do not believe conveying predictions about recovery will have negative consequences, e.g., due to the impact of uncertain predictions later proving incorrect, or predictions about a negative outcome reducing patient motivation and/ or impacting mood | Some professionals believe there are negative consequences to providing information about recovery, due to the uncertainty of stroke recovery and the impact if these later transpire to be false (negative emotional reactions from patients and families, possibility of complaints), or when the outlook is suboptimal and there is potential for long-term disability, and the potential impact this may have on patients' mood and subsequent motivation to engage with therapy | Education, Persuasion, Modelling, Enablement | Explain the benefits of providing information about recovery to patients and families if provided sensitively and compassionately, e.g., to support adjustment, enable planning, but steps should be taken to ascertain how much/ the type of information they want to receive and to convey uncertainty (**Information about social and environmental consequences**) Demonstrate how to convey uncertainty when providing information about recovery and how to manage patients' and families' emotional responses when they occur (**Demonstration of the behaviour**) Demonstrate how to provide information about recovery in positive ways, to foster hope and motivation (**Demonstration of the behaviour**) Present a speech by an expert (stroke survivor or carer) outlining the benefits of providing information about recovery, even where it might be uncertain or involve 'bad news' (**Credible source**) Advise the professional to list and compare the advantages and disadvantages of providing information about recovery (**Pros and cons**) |
| | Intentions | N/A | N/A | N/A | N/A |
| | Goals | N/A | N/A | N/A | N/A |
| **Automatic motivation** | Reinforcement | N/A | N/A | N/A | N/A |
| | Emotion | Professionals are able to manage their own emotions in relation to difficult conversations about recovery, including anxiety and distress | Some professionals feel anxious about approaching conversations about recovery, particularly when breaking bad news, or experience distress following these conversations. | Persuasion, Enablement | Advise on the use of stress management skills to reduce anxiety (**Reduce negative emotions**) Arrange emotional support from within the MDT or from service managers to support professionals prior to difficult conversations (**Social support (emotional)**) Advise professionals to share responsibility/ approach difficult conversations alongside colleagues where possible/ appropriate (**Conserve mental resources**) Normalise the negative emotions experienced by professionals following difficult conversations (**Information about emotional consequences**) |

BCTs = Behaviour Change Techniques

intervention and work was required to select those most likely to be feasible and acceptable to intervention recipients; we thus presented them to stroke unit professionals in a survey study.

## Phase 2: Online survey study

Forty-eight stroke unit professionals participated, representing a range of MDT professionals (see Table 3). Most were physiotherapists (n = 16; 33%), speech and language therapists (n = 13; 27%), occupational therapists (n = 5; 10%) and nurses (n = 5; 10%), with smaller numbers of doctors (n = 2;4%), clinical psychologists (n = 2; 4%), orthoptists (n = 2; 4%), dietitians (n = 2; 4%) and a patient mentor (n = 1, 2%). Most were female (n = 43; 90%), over 30 years old (n = 43; 90%) and had worked in stroke care for more than ten years (n = 31; 65%).

All barriers received a median importance rating of 4 (very) or 5 (extremely important to address within an intervention), with at least 70% of participants rating each as either very or extremely important to address (range = 70–94%). Each individual BCT was rated as either likely or very likely to be useful by a mean of 88% of participants (range = 60–100%), and likely or very likely to be feasible by 89% (range = 52–98%).

Although we aimed to identify the five most important barriers to address in an intervention, there was a tie between those ranked fifth and sixth, therefore both were included. The highest-ranking barriers concerned perceived: lack of confidence (94%); insufficient communication skills (92%); lack of knowledge of the benefits (85%); difficulties in deciding when and in what format to provide information (83%); absence of private and quiet spaces for discussions (81%); and lack of generic written information to support conversations (81%). These barriers were related to the COM-B components (and linked TDF domains) of: Psychological

Table 3. Participant demographics.

|  | n (%) |
|---|---|
| **Female** | 43 (90) |
| **Age group (%)** | |
| • **18–30 years** | 5 (10) |
| • **31–40 years** | 12 (25) |
| • **41–50 years** | 18 (37.5) |
| • **51–60 years** | 13 (27) |
| **Professional background** | |
| • **Physiotherapist** | 16 (33) |
| • **Occupational Therapist** | 5 (10) |
| • **Speech & Language Therapist** | 13 (27) |
| • **Nurse** | 5 (10) |
| • **Doctor** | 2 (4) |
| • **Orthoptist** | 2 (4) |
| • **Clinical Psychologist** | 2 (4) |
| • **Dietitian** | 2 (4) |
| • **Patient Mentor** | 1 (2) |
| **Years of experience in stroke care** | |
| • **<1 year** | 3 (6) |
| • **1–5 years** | 8 (17) |
| • **6–10 years** | 6 (12.5) |
| • **>10 years** | 31 (65) |

(N = 48)

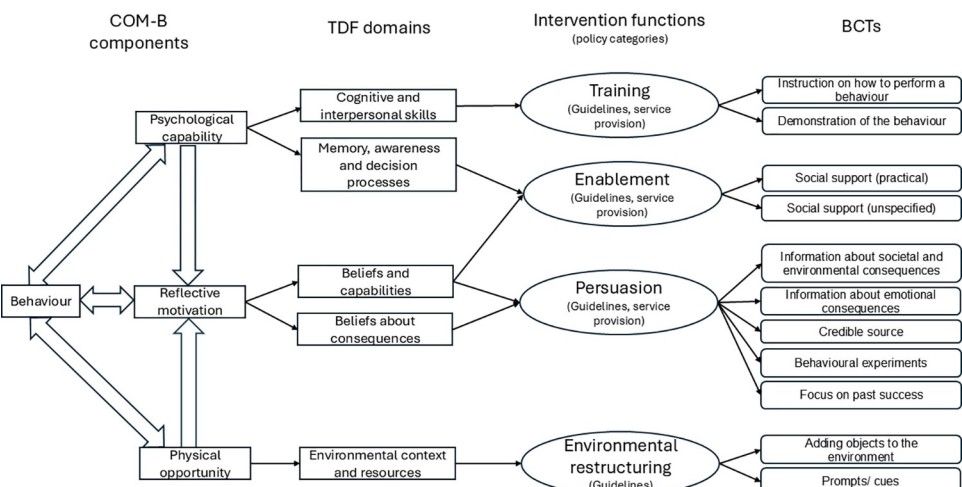

**Fig 2. Selected COM-B components, TDF domains, Intervention Functions and BCTs.** COM-B = Capability, Opportunity, Motivation model of Behaviour; TDF = Theoretical Domains Framework; BCTs = Behaviour Change Techniques.

Capability (Cognitive and interpersonal skills; Memory, awareness and decision processes), Reflective Motivation (Beliefs about capabilities; Beliefs about consequences) and Physical Opportunity (Environmental context and resources); see Fig 2. The 20 BCTs which were ranked most likely to be useful were selected from the 24 proposed to address these barriers. We retained those most likely to be feasible (where ≥90% rated them as likely or very likely to be feasible), resulting in the selection of twelve BCTs for intervention inclusion (see Table 4); these related to the Intervention Functions of Training, Enablement, Persuasion, and Environmental restructuring. They encompassed techniques that could be used to enhance the skills and increase the confidence of stroke unit professionals when providing recovery information, strategies to persuade them of the benefits, and approaches to adapt the physical environment to facilitate information provision.

Whilst respondents' ratings suggested most of the selected BCTs would be feasible, their qualitative feedback provided insights into their perspectives in terms of mode of intervention delivery. In terms of overall feasibility, the most cited barrier to the suggested BCTs in our survey was a lack of staff time and capacity. Respondents however suggested a range of ways that the selected BCTs could be feasibly delivered, including making use of technology (e.g., online presentations, written resources, and discussion groups) and existing ward processes and resources, such as in-service training sessions and the experience of colleagues. Therefore, options for intervention delivery will include face-to-face and in a group setting, representing greater value for money and more efficient use of intervention deliverers' time than an individually targeted intervention. There is potential however for some BCTs to be delivered at a distance, either individually or in groups, e.g., using webinars for group delivery or online materials such as videos that could be accessed individually.

## Discussion

The BCW approach incorporating the TDF was effective in supporting early systematic and theory-driven development of an intervention designed to improve conversations about post-stroke recovery. This approach facilitated development of understanding of the problem in behavioural terms to enable identification of Intervention Functions and BCTs that may be

**Table 4. Suggested core components of a professional-focused intervention to improve provision of information about recovery.**

| Selected intervention functions | COM-B components (linked TDF domains) served by Intervention Functions | Selected BCTs to deliver intervention functions | Intervention components |
|---|---|---|---|
| **Training (imparting skills)** | Psychological capability (Cognitive and interpersonal skills): Increasing professionals' skills, confidence, and comfort in providing information | Instruction on how to perform a behaviour Demonstration of the behaviour | Advise professionals on how to decide when and in what format they should provide information to individual patients/ families Demonstrate to professionals: • how they should deliver information about recovery sensitively and compassionately • examples of conversations occurring in different ways (e.g., at different times, supported by written documentation or not) with patients with different needs |
| **Enablement (increasing means/ reducing barriers to increase capability or opportunity)** | Psychological capability (Memory, awareness and decision processes) and Reflective motivation (Beliefs about capabilities): Increasing professionals' capability and confidence through encouraging team-working and support | Social support (practical) Social support (unspecified) | Advise professionals on how they should request support from colleagues across the MDT when making decisions about when and in what format to provide information Encourage professionals to provide support and encourage their colleagues when they have had discussions with patients and families about recovery |
| **Persuasion (using communication to induce positive or negative feelings or stimulate action)** | Reflective motivation (Beliefs about capabilities and Beliefs about consequences): Increasing professionals' understanding about patients' and families' information needs and the importance of meeting them, and encouraging self-reflection to increase their confidence in their own capabilities | Information about societal and environmental consequences Information about emotional consequences Credible source Focus on past success Behavioural experiments | Provide information to professionals about patients' and carers' information needs about recovery from established literature Provide information to professionals about the emotional consequences for patients and carers if information about recovery is not provided effectively Present a speech by an expert (stroke survivor/ carer) to professionals outlining the known benefits and risks to providing information about recovery Encourage professionals to think about times they have successfully shared information with patients and families and information was well-received Ask professionals to try providing information about recovery (after structured training and as part of supervised practice) and to note patients' and families' reactions |
| **Environmental restructuring (changing the physical or social context)** | Physical opportunity (Environmental context and resources): Providing physical resources to support professionals during recovery conversations and instructions to ensure they are readily available | Adding objects to the environment Prompts/ cues | Provide professionals with (or support them to identify) generic written information to provide to patients/ families Ensure written information is readily available in a specific location for professionals to access when required |

effective in changing professionals' behaviour to facilitate provision of information. We incorporated stakeholders' experiences and views throughout the process of early intervention development, through using previously collected qualitative data from patients with stroke, carers, and stroke unit staff to support identification of barriers and strategies to address them, and specifically collected data from staff via an online survey to prioritise these barriers and select the strategies mostly likely to meet the needs of those whose experiences we wish to improve. Survey results led us to focus our intervention around addressing barriers in five TDF domains: Cognitive and interpersonal skills (lack of perceived skills in providing information), Memory, awareness and decision processes (deciding when and in what format to provide information); Beliefs about capabilities (confidence in providing information); Beliefs about consequences (lack of awareness of the benefits of providing information and the impact

of not providing it) and Environmental context and resources (an absence of available written resources to support conversations). The proposed intervention strategy included 11 BCTs relating to the Intervention Functions of Training (e.g., demonstrating how information could be provided sensitively to patients with different needs), Enablement (providing advice to seek and provide support to/from colleagues), Persuasion (including providing information to professionals about patients' information needs and the consequences of not providing information, and Environmental restructuring (providing professionals with general written information to provide to patients/ families and ensuring this is readily available). Further work is underway to build on this early work using coproduction to ensure that the intervention strategy and materials are feasible and acceptable to both professionals (as information providers) and patients and carers (as information recipients).

The use of the BCW and TDF in intervention development has facilitated the consideration of a wide range of factors with the potential to impact professionals' behaviour in relation to provision of post-stroke recovery information. Previously reported interventions designed to address communication skills (including breaking bad news) infrequently use theory in their development and thus typically focus only on communication skills training techniques such as didactic teaching and simulation with feedback on performance [18]. These techniques have been demonstrated to improve proximal outcomes such as clinicians' self-rated confidence and observer-rated performance [34]; our study adds to this literature by using theory to understand the mechanisms through which these interventions could lead to such outcomes. However, the extent to which these interventions lead to sustained changes in clinical practice and impact on more distal outcomes such as patient and carer experience are rarely studied; it is typically assumed that improvements to clinicians' confidence and performance lead to enhanced outcomes [35, 36]. Whilst professionals' skills and confidence were highlighted as important to address in this study, the use of the BCW approach, and the TDF, has supported exploration of a range of other important barriers, which can inhibit professionals from sharing of information about prognosis or recovery, e.g., the physical environment. Addressing these barriers may prove more effective in changing practice and enable professionals to implement their new skills. Future research should however examine the impact of such interventions on the experiences of patients and carers, as well as professionals' confidence and skills.

Application of the BCW approach provided us with a route to action guidance from the MRC in developing complex interventions, using theory from the start of development to describe how an intervention is expected to function and lead to its intended effects [19]. This will be refined as intervention development progresses. In addition, the structured and systematic nature of the BCW approach facilitated transparent and detailed reporting of the intervention development process, which is commonly absent in the development of stroke rehabilitation complex interventions [37]. Such reporting is needed to develop understanding of the results of subsequent efficacy studies, particularly where they fail to demonstrate significant effects [20].

Both the MRC framework [19] and the Opportunity component of the COM-B model in the BCW approach [22] encouraged us to consider the context in which the intervention will be delivered. The qualitative data informing this study were collected in the stroke unit context, which facilitated identification of the potential barriers to conversations about recovery. However, stroke units vary widely in their physical environments, management, and processes, such that some barriers might be vastly important in some, and not others. For example, in some units, where private spaces are available and used, addressing barriers related to Physical Opportunity may have little impact. The MRC framework highlights the potential for interventions to be effective in some contexts and ineffective, or even harmful in others,

accentuating the importance of flexibility [19]. The BCW approach has enabled the delineation of the intervention into component parts; those considered core to delivery will be required, whilst permissible variation will be agreed with intervention deliverers to permit flexibility to different contexts. Refinement of the intervention (through our ongoing coproduction work and subsequent feasibility testing) will further shape the intervention and help to identify the content which is considered 'core' in all settings, and which will be employed in specific contexts; and how implementation may vary. For example, provision of information in the early stages post-stroke (where the focus is on survival) is likely to differ (e.g., in its uncertainty and its urgency) from that which is provided in long-stay rehabilitation units (which typically focus on improving quality of life with on-going disability). Additionally, it may be identified that specific staff groups require different training recognising their existing skills (e.g., doctors) or the expectations and remit of their role in information provision (e.g., therapy assistants are unlikely to have a direct role in providing information but may need training to ensure that their approach to conversations about recovery is in line with the information provided by other professionals). Delivering such training to different staff groups and settings may require different approaches, e.g., utilising online options, as suggested by survey participants.

Like other researchers [38, 39], we found that using the BCW approach to intervention development felt at times open to the subjective judgements of the research team. For example, we decided that professionals' behaviour would be the target of the intervention, a decision which then informed subsequent development. Although this judgement was reasoned and subject to agreement by our team, should different decisions have been made, e.g., focusing on the behaviour of patients and/ or carers, or all three groups, the intervention function and BCTs are likely to have been different. Additionally, the aim of the intervention was focused on changing professionals' behaviour, rather than driving change at the organisational level. It may be that organisational change would be required to facilitate a cultural shift to a mode of working where information is readily shared and available for patients and carers, such that they feel empowered to request it.

Although our research team included experienced researchers from several multidisciplinary clinical backgrounds, to address the potential subjectivity of applying judgements and encouraged by the MRC framework [19] to consider intervention implementation throughout the development process, we chose to consult stakeholders using an online survey to help inform decision-making. Other studies have used the BCW approach to underpin intervention development using participatory methods, such as coproduction, enabling the involvement of stakeholders throughout the process [40, 41]. As well as facilitating decision-making, such involvement may help to facilitate later implementation and uptake, ensuring that intervention strategies are feasible and gaining buy-in from intervention deliverers and/ or (in this case) those whose behaviour is to be targeted.

Although Michie et al. recommend the use of qualitative data collection to develop understanding of the target behaviour [22], this may be limited to specific contexts and although triangulation of data from different sources is recommended to consolidate understanding, the extent to which findings are transferable to other contexts may be limited. In our study, although we generated rich and detailed data, and triangulated data collected from different sources (patients, carers, and professionals) and via different methods (observations, interviews, documents), the qualitative work which informed our application of the BCW [16] was conducted in only two stroke units in one English county. It was thus difficult to assess whether the barriers we identified were transferable to stroke units more widely. Our survey study validated these findings as important barriers to discussions about recovery across the UK, which gives us confidence that our final intervention will be useful for others in these

contexts, as well as highlighting the need for interventions of this type. We prioritised the six highest ranking barriers to address within our intervention strategy because we considered that the proposed intervention needed to pay sufficient attention to a barrier if it were to effectively address concerns about the potential complexity and implementation challenges of the developed intervention. However, it is noteworthy that all twelve of the barriers we presented in our survey were considered important or very important to address within an intervention by a majority of participants (>70%). Our continuing development remains mindful of the other barriers, and we will consider if they can be incorporated into the intervention without diverting attention from the most important barriers. Future evaluation will need to examine whether including the remaining barriers substantially hampers provision of recovery information if and when the targeted barriers are included.

Some limitations within our survey study must however be acknowledged. The use of convenience sampling rendered us unable to control who viewed the advert and completed the survey; it is therefore unclear whether specific characteristics influenced completion. Although we did not aim to generate generalisable findings, but rather to gain understanding of a wider perspective on the topic, it may be that participants' views are not representative of stroke unit professionals in general. In particular, some professional groups, e.g., OTs and nurses, were under-represented. Although the sample size was relatively small, we did however generate views from a wide range of professionals, including more peripheral MDT members, such as psychologists, orthoptists, and dietitians. This suggests there is interest in this topic outside of the core MDT and behaviour change may be possible in these groups. Our sample also included many experienced professionals; over half had more than ten years of experience in stroke care. This may have resulted in the sharing of helpful knowledge about effective strategies to improve conversations about recovery based on past experiences, however perspectives of those with less experience (and therefore potentially more likely to benefit from an intervention) may have been missed. Finally, although the survey was open to professionals across the UK and widely advertised, data on participants' location were not collected, and it is therefore possible that responses reflected views of professionals at only a small number of stroke units or within a specific region.

## Conclusions

Use of the BCW approach (informed by previously collected qualitative data) successfully structured early development of an intervention to improve conversations about recovery on the stroke unit, facilitating understanding of professionals' behaviour relating to information provision, and the barriers they may experience. The approach enabled us to select Intervention Functions and BCTs that could be effective in changing these behaviours, however this approach frequently relies on subjective judgements made by researchers. To overcome these challenges and to incorporate stakeholder involvement as recommended in the MRC guidelines, we sought the views of stroke professionals via an online survey. This provided validation of the barriers identified through smaller scale qualitative work, and enabled us to gain insight into professionals' perceptions of those felt most important to address within an intervention, and the BCTs with most potential to be useful and clinically feasible. This novel approach could be used by other researchers in the field of complex intervention development. Our current coproduction work will develop intervention materials; the result will be a theory-driven intervention designed to support the development of professionals' skills and confidence to discuss recovery in ways which meet the needs of patients, carers, and professionals. Future research is however required to assess whether the use of theory and stakeholder engagement has facilitated the creation of an intervention which is effective in improving professionals'

skills and confidence, and patients' satisfaction with information, engagement in shared decision-making and adjustment.

## Supporting information

**S1 File. Completed GUIDED checklist.**
(PDF)

## Acknowledgments

The authors wish to thank Dr Jennifer Hall for providing advice on use of the BCW and reviewing, and making suggestions on, an early draft of the proposed BCTs.

## Author Contributions

**Conceptualization:** Louisa-Jane Burton, Anne Forster, Judith Johnson, Thomas F. Crocker, David J. Clarke.

**Formal analysis:** Louisa-Jane Burton, Anne Forster, Thomas F. Crocker, David J. Clarke.

**Funding acquisition:** Louisa-Jane Burton, Anne Forster, David J. Clarke.

**Investigation:** Louisa-Jane Burton.

**Methodology:** Louisa-Jane Burton, Anne Forster, Judith Johnson, Thomas F. Crocker, David J. Clarke.

**Project administration:** Louisa-Jane Burton.

**Supervision:** Anne Forster, Judith Johnson, Thomas F. Crocker, David J. Clarke.

**Writing – original draft:** Louisa-Jane Burton.

**Writing – review & editing:** Anne Forster, Judith Johnson, Thomas F. Crocker, David J. Clarke.

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
