## [Decision Letter · Decision Letter 0]

18 Oct 2024

PONE-D-24-27432Using the Behaviour Change Wheel to develop an intervention to improve conversations about recovery on the stroke unitPLOS ONE

Dear Dr. Burton,

Thank you for submitting your manuscript to PLOS ONE. After careful consideration, we feel that it has merit but does not fully meet PLOS ONE’s publication criteria as it currently stands. Therefore, we invite you to submit a revised version of the manuscript that addresses the points raised during the review process.

We look forward to receiving your revised manuscript.

Kind regards,

Lesley Smith, PhD

Academic Editor

PLOS ONE

Journal Requirements:

2. Thank you for stating the following financial disclosure: Funding for this work was provided by the Stroke Association, via its Postgraduate Fellowship programme [ref: TSA PGF 2017-02].   

5. Your abstract cannot contain citations. Please only include citations in the body text of the manuscript, and ensure that they remain in ascending numerical order on first mention.

Additional Editor Comments:

The manuscript reports on an important topic using gold-standard methodological frameworks. It is well written on the whole and easy to follow, however there are areas where more or clearer information is required. These are outlined below:

Overall, throughout the manuscript, the relationship between BCW, COM-B and TDF and how they were applied is not clear. A diagram illustrating this relationship might help show this more clearly. Presumably it was an iterative process?

Abstract conclusion needs revising. It should more accurately reflect your findings rather than stating what you did. Perhaps phrasing it along the lines of "the application of the COM-B (or is it BCW?) to intervention development for......was successful......further research will......

The conclusion should match the conclusion in the main body of the manuscript.

Background

The aim of the work needs stating. You may also consider stating some objectives which could link to the different phases of the research. This would help structure the methods and results.

Methods

There is a lack of consistency in your use of COM-B and BCW. COM-B model applied in first paragraph, but then in phase 1 it is BCW. Also in your use of terms such as multi-phase, then you mention phase 1. Be very clear how many phases there were in the first paragraph and briefly describe them before going into the details in subsequent paragraphs.

There is scant information on how the TDF was applied - please expand on this. Why was it used?

More detail is required on how APEASE was judged or applied?

The rationale for only shortlisting 5/6 barriers is required.

Results

How were the barriers mapped to the TDF domains?

47 stakeholders in the text but 48 in the table.

You selected 6 barriers but the results indicated that all were very or extremely important to address. This needs further explanation and the implications of this need picking up in the discussion.

Table 4 describes intervention components but how these were specified and selected needs explanation.

Discussion

The initial paragraph should answer the extent to which you addressed your aim rather than re-stating your aim. The first sentence could be broken into 2 to make a clearer message.

Please clarify if the selection of barriers was informed by the survey rather than previous work. Previous work identified them but you selected the ones to address in the survey?

The discussion would benefit from re-structuring so that you clearly describe what you found first before you make comparisons with the wider literature.

Line 395 - rather than describing the MRC framework and BCW again discuss how these facilitated your research aims and objectives. Discuss the application of it rather than rationale for it's use.

Expand on your statement that the final intervention will be flexible - what does this mean and how will it be adaptable to other settings, populations etc.

line 380-382 - this belongs in the introduction/background section as it provides a rationale for your study.

Conclusion

This should align with the abstract conclusion and needs improving. You used frameworks and processes for intervention development. To what extent did these enable you to achieve your aims, any limitations and what comes next.

Reviewers' comments:

Reviewer's Responses to Questions

**Comments to the Author**

1. Is the manuscript technically sound, and do the data support the conclusions?

Reviewer #1: Yes

2. Has the statistical analysis been performed appropriately and rigorously? 

Reviewer #1: N/A

3. Have the authors made all data underlying the findings in their manuscript fully available?

Reviewer #1: Yes

4. Is the manuscript presented in an intelligible fashion and written in standard English?

Reviewer #1: Yes

5. Review Comments to the Author

Reviewer #1: This is an interesting manuscript and although I am from a different discipline, it was exciting to consider how this approach could be applied across different areas of healthcare.

Please find below a couple of minor comments to consider which could be clarified further to enhance the readability / improve the accessibility and add to the interest of the paper. Some readers may be reading your paper for your methodological approach rather than topic itself, so it is good to consider this too.

Maybe consider providing a little more detail about the TDF. This would help to show the reader why you used both the TDF and COM-B, rather than one or the other.

Overall, I felt that you have explained quiet a complicated methodology well and the tables do help to clarify the different aspects and show your rational and explanations for each. It did take some time to understand and follow all of the stages, so if there was a way to reduce / make any sections more concise, this would help the overall readability, but I understand that this may not be possible to do without losing the required level of detail.

6. PLOS authors have the option to publish the peer review history of their article (what does this mean?). If published, this will include your full peer review and any attached files.

Reviewer #1: No

---

## [Author Response · Author response to Decision Letter 0]

8 Dec 2024

We thank the editor and reviewers for their comments. Our response and changes made have been detailed in the attached Response to Reviewers.

---

## [Editor Report · Decision Letter 1]

22 Dec 2024

Using the Behaviour Change Wheel to develop an intervention to improve conversations about recovery on the stroke unit

PONE-D-24-27432R1

Dear Dr. Burton,

We’re pleased to inform you that your manuscript has been judged scientifically suitable for publication and will be formally accepted for publication once it meets all outstanding technical requirements.

Kind regards,

Lesley Smith, PhD

Academic Editor

PLOS ONE

Additional Editor Comments (optional):

You have revised the manuscript by addressing the peer review comments and have managed to describe your methodological processes clearly and comprehensively. I think the amendments have improved the manuscript, and it is ready to be published. Well done.
---

## [Editor Report · Acceptance letter]

26 Dec 2024

PONE-D-24-27432R1 

PLOS ONE

Dear Dr. Burton, 

I'm pleased to inform you that your manuscript has been deemed suitable for publication in PLOS ONE. Congratulations! Your manuscript is now being handed over to our production team.

Kind regards, 

on behalf of

Professor Lesley Smith 

Academic Editor

PLOS ONE